# Plotter Cut Stencil Masks for the Deposition of Organic and Inorganic Materials and a New Rapid, Cost Effective Technique for Antimicrobial Evaluations

**DOI:** 10.3390/mi14010014

**Published:** 2022-12-21

**Authors:** Andre Childs, Jorge Pereira, Charles M. Didier, Aliyah Baksh, Isaac Johnson, Jorge Manrique Castro, Edwin Davidson, Swadeshmukul Santra, Swaminathan Rajaraman

**Affiliations:** 1Department of Material Science and Engineering, University of Central Florida, Orlando, FL 32816, USA; 2Department of Chemistry, University of Central Florida, Orlando, FL 32816, USA; 3Burnett School of Biomedical Sciences, University of Central Florida, Orlando, FL 32827, USA; 4Department of Mechanical and Aerospace Engineering, University of Central Florida, Orlando, FL 32816, USA; 5Department of Electrical and Computer Engineering, University of Central Florida, Orlando, FL 32816, USA; 6NanoScience Technology Center, University of Central Florida, Orlando, FL 32826, USA

**Keywords:** plotter cutter, stencil mask, material patterning, Kirby-Bauer, antibiotics testing, minimal inhibitory concentration detection

## Abstract

Plotter cutters in stencil mask prototyping are underutilized but have several advantages over traditional MEMS techniques. In this paper we investigate the use of a conventional plotter cutter as a highly effective benchtop tool for the rapid prototyping of stencil masks in the sub-250 μm range and characterize patterned layers of organic/inorganic materials. Furthermore, we show a new diagnostic monitoring application for use in healthcare, and a potential replacement of the Standard Kirby-Bauer Diffusion Antibiotic Resistance tests was developed and tested on both *Escherichia coli* and *Xanthomonas alfalfae* as pathogens with Oxytetracycline, Streptomycin and Kanamycin. We show that the reduction in area required for the minimum inhibitory concentration tests; allow for three times the number of tests to be performed within the same nutrient agar Petri dish, demonstrated both theoretically and experimentally resulting in correlations of R ≈ 0.96 and 0.985, respectively for both pathogens.

## 1. Introduction

Traditionally MEMS stencils have been largely fabricated using laser micromachining, computer numerical control (CNC) milling, deep reactive-ion etching (DRIE), and other photolithographic approaches due to their great precision and ease of workflow integration achieving excellent feature resolutions down to 10’s of nanometers [1,2,3,4,5,6,7]. However, with lengthier processing times, high cost for prototyping, and the requirements of a cleanroom facility, these technologies have become much less appealing lately particularly, due to accessibility concerns for low-resource settings and low-income countries [8,9,10,11]. With extreme process optimization, technologies such as plotter cutters and other electronic cutters could replace older stencil masking technologies [12]. As these tools have been shown to be more effective, especially in microfluidic and biosensor applications in recent years and are extremely appealing to the growing makerspace and makerspace microfabrication communities [10,13,14,15,16].

Previous work has demonstrated that these current benchtop plotter cutters can reach down to ~200 μm in minimum feature sizes and according to Bartholomeusz et al. most plotter cutters have a theoretical resolution of ~25 μm which is harder to demonstrate repeatably. The plotter cutter used by Bartholomeusz et al. was a FC5100A-75 from Graphtec (~$4000 in cost), demonstrated excellent features. With this tool microchannels of ~38 μm in width and serpentine patterns of 78 ± 23 μm in dimensions, with poor accuracies up to ~26% in discrepancy were defined [17]. The recent work by Islam et al., measured the accuracy of four different microfluidic patterns: straight lines, serpentines, zig zag patterns and square microchannels for electrophoresis, mixing, particles separation and bubble transport, respectively. They reported error values in the accuracy of the cuts were as high 26.5% and as low as 9.09% with the smallest cut being 208.07 ± 9.09 μm in a double-sided pressure sensitive adhesive. The plotter cutter used in this work was a Graphtec CE6000-40 (~$2000 in cost). Additionally work from Yuen and Goral showed a 200 μm limit for serpentine channels using a QuicKutz ^®^ Silhouette™ SD (~$300 in cost) [18]. All these results are summarized in Appendix A.

In this work, a similar Silhouette™ Plotter Cutter Cameo 4 (SPC4) which is relatively inexpensive (~$300 in cost) is utilized. The SPC4 can cut up to 3 mm thick materials, has two carriages: one that can produce up to 210 gf (2.01 N) force and the other up to 5 kgf (200 N). The 5 kgf force is the highest reported force in this machine class and has a work envelope of approximately 30.48 cm by 30.48 cm cutting mat. The blade that is commonly used for the SPC4 Silhouette™ Plotter Cutter is the Autoblade, which can be fitted into one of the carriers of the system. Additionally, three more blades can be retrofitted in a second carrier in the system: a Kraft blade, a Rotary blade, and a Punch tool. Researchers have typically worked with the SPC4 using the kraft cutter with an engraving tip width of 100–300 μm, as this tool has produced channel resolutions widths of 45 ± 5 μm and 50–300 μm in depth [19,20]. However, to our knowledge, no one has done extensive research on the Autoblade SPC4.

Even though the utilization of plotter cutters to create custom stencils is not new, plenty of new research exploration in terms of tools and materials for cutting, as well as applications remain. In this paper, we design and microfabricate stencil masks with Kapton^®^ 300 HN due to its low cost, favorable mechanical and electrical properties, flexibility, use as electrical insulation, chemical/radiation resistance, and ability to operate over wide temperature ranges (−269 to 400 °C). Other advantages of Kapton^®^ for this application include a tensile strength of 24,000 MPa, shrinkage at 200 °C of just 0.35%, moisture absorption (4.0%), excellent dielectric strength (4500 kV/mm), volume resistivity (10^12^ ohm/cm), dielectric constant (3.5) and dissipation factor of just 0.0036 at 1 kHz [21].

Kapton^®^ films have been used in cardiovascular applications [22], space station solar cell arrays [23], and flexible electronic sensors [24,25,26]. Kapton^®^ films have also been used as stencil masks by our group and other researchers [15,26,27,28]. Specifically, the usage of these microfabricated stencil masks can be extrapolated to a myriad of applications such as Interdigitated Electrodes (IDEs) for impedance measurement, dielectric layer deposition, and microenvironment patterning for the manipulation and selective localization of bacterial and eukaryotic cells. Such localization is imperative to the study of cellular behavior such as morphology, and response to pharmaceutical drugs in critical disease states such as cancer [29,30]. With the rise of antibiotic resistant pathogens, it has become important to quickly assess the vulnerability of bacteria to different antibiotics to better determine medical treatment in humans and animals or crop disease management in plants. The Kirby-Bauer Disk Diffusion Test (DDT) allows for antimicrobial assessment of different compounds against bacterial population [31,32] that require large test area (10 s of mm^2^) and sample volume (20 µL or more), limiting device use efficiency.

This work proposes a novel variation of the DDT that utilizes Kapton^®^ stencils to minimize and confine the area required for testing. Due to the importance of this assessment for both the biomedical and agricultural fields the proposed test was assessed against a strain of Escherichia coli (*E. coli* K-12), a well known model human pathogen that can be transmitted from animal to human, and *Xanthomonas alfalfae* (*X. alfalfae*), a model pathogen of the Xanthomonas family, which is known to cause significant damage to crops, resulting in high yield loss [33,34,35,36].

The capabilities demonstrated in this paper showcase how the SPC4 tool provides precision with a multitude of swift cuts leading to the rapid prototyping of material layers for accurate diagnosis, analysis treatment or acute action. We analyze the best cutting parameters for 300 HN Kapton^®^ and the drag method with force penetration using the tetragonally shaped Autoblade. We additionally utilized the stencil masks, for varied deposition of organic and inorganic materials such as Cr, Au, SiO2, and gelatin.

## 2. Materials and Methods

### 2.1. Methods of Imaging

The following tools were used for imaging all the reported data in the results and Appendix A sections. The cutting surfaces and materials casting were analyzed using the Confocal Microscope (Keyence BZ-X800, Itasca, IL, USA), the Dektak Xt profilometer (Billerica, MA, USA), the Zeiss Ultra 55 Scanning Electron Microscope (SEM) (Oberkochen, Germany) and Apple iPhone 11 Pro (Cupertino, CA, USA).

### 2.2. Analysis of the Cutting Blade

The cutting blade from the Silhouette™ Plotter Cutter Cameo 4 (Autoblade) was removed from the cutter for analysis. The blade angle and the different portions of the cutting surface were analyzed thoroughly from the surface of the material, cutting blade edge dimensions, angles, etc., with the Confocal Microscope Keyence BZ-X800. Force measurements to determine the cutting forces were performed using a 400 series square Force Sensing Resistor (FSR) from Adafruit (New York, NY, USA). In order to elucidate the force required to cut through 2 layers Kapton^®^ substrate, a Force Sensitive Resistor (FSR, Adafruit) was placed in between the substrate and the cutting mat. Straight line channels (Length: 12.7 mm × Width: 2 mm) were machined and the resultant forces measured using the FSR.

### 2.3. Stencil Mask Designs

Kapton^®^ Films 300HN were cut using the Silhouette™ Cameo 4 plotter cutter (Figure 1A) with feature dimensions of widths 2 mm, 1 mm, 750 μm, 500 μm, 250 μm, 200 μm with lengths of 12.7 mm and 10 mm, for the “Straight Line Channels” and “Square Channels” designs, respectively. Both were designed using SOLIDWORKS (Dassault Systèmes, Paris, France). The “Serpentine Channels” were designed in SOLIDWORKS as well with feature dimensions: 4 mm, 2 mm, 1500 μm, 1000 μm and 500 μm with channel widths of 2 mm and length of 5 mm. The geometries were chosen due to the common use in many microfluidic devices and other microsystems. For instance, the straight channels are used in electrophoresis techniques, the serpentine channels for particles separation and the square channels for bubble transport [12]. A Metal Cr (Chromium, 50 nm) was deposited through these cut stencils to demonstrate conductive layer depositions.

### 2.4. Gelatin Casting

A 10 wt% gelatin solution was prepared using Gelatin from Bovine Skin (Powder) purchased from Sigma Aldrich (St. Louis, MO, USA). Kapton^®^ stencil masks were attached to glass slides using Kapton^®^ tape and then the gelatin was cast through the following stencil masks “Straight Line Channels”, “Square Channels”, “Serpentine Channels” on the glass slides using transfer pipettes. (Figure 1B).

### 2.5. Interdigitated Electrode Design (IDE) and Microfabrication

IDEs of design dimensions, 1 mm and 2 mm in width, 5 mm and 2.5 mm pitch were developed using SOLIDWORKS. These designs were additionally cut using the Silhouette Cameo 4 Plotter Cutter for the study of rapid microfabricated Interdigitated Electrodes. These Kapton^®^ masks were attached to glass slides as described in Section 2.4, and Cr/Au (50/25 nm) was deposited using a Temescal E-beam evaporator (Ferrotec, Livermore, CA, USA) with a chamber pressure of $8\times 10^{-6}$ torr at a deposition rate of 1.5 Å/s (Figure 1C).

### 2.6. Silicon Dioxide Deposition

Kapton^®^ stencil masks with “Straight Line Channels”, “Square Channels”, “Serpentine Channels” were developed as described in Section 2.3 and placed on glass slides such as in Section 2.4. Silicon dioxide deposition was performed with an Orion Trion Plasma Enhanced Chemical Vapor Deposition (PECVD) (Plasma Therm, Clearwater, FL, USA) at a rate of 50 nm/min for 10 min at 55 °C and 300 °C using TEOS and O_3_ (Airgas, Radno, PA, USA). Analysis of the deposited SiO_2_ layers was performed with the Dektak Xt profilometer and Zeiss SEM (Figure 1D).

### 2.7. Bacteria Handling

Bacteria stocks were stored at −80 °C in a Thermo Scientific REVCO Freezer ULT2186-5-AVA (Waltham, MA, USA). For the tests, bacteria were streaked onto Mueller Hinton II agar (MHA) (Becton Dickinson and Company, Franklin Lakes, NJ, USA) plates and incubated for two days. Afterward, individual colonies were utilized to inoculate 10 mL of nutrient broth and set on an incubator shaker at 150 rpm for 24 h before being utilized for the antimicrobial assessment. *E. coli* K-12 was incubated at 37 °C, while *X. alfalfae* was incubated at 28 °C.

### 2.8. Disk Diffusion Assay

The disk diffusion assay was performed according to a previously published protocol with some variations [31]. Briefly, 6.0 mm disks were imbibed in 20 μL of an antibiotic solution to deposit a defined mass of the compound. Separately, MHA plates were streaked with a sterile cotton applicator soaked in a bacterial solution set to a bacterial density of ~10^8^ colony forming units per milliliter (CFU/mL). Once the agar plates were dry, the antibiotic discs were placed separately on the surface of the agar, the plate was then closed and sealed using parafilm. Before measuring the radius of inhibition *E. coli* K-12 and *X. alfalfae*, inoculated tests were incubated for 24 h at 37 °C and 48 h at 28 °C, respectively in a Fisher Scientific Isotemp Incubator (Model No: 637F). For the templated Stencil Disk Diffusion test (SDDT), MHA agar plates were covered with the Kapton^®^ stencil masks (design dimensions: 6 mm diameter circle and a linear channel of 42 mm in length and 2 mm in width, Figure 1E) before streaking the bacteria only on the exposed agar surface. Antibiotics Streptomycin Sulfate (2.5–20 μg), Oxytetracycline Hydrochloride (7.5–60 μg), and Kanamycin Sulfate (7.5–60 μg) were utilized in the new diffusion test increasing doses linearly. All tests and characterizations were performed in triplicates (i.e., n = 3).

## 3. Results

### 3.1. Analysis of the Cutting Blade

The key cutting feature of the plotter cutter blade is shown in Figure 2. The Autoblade has an ellipsoidal curvature of length ~1400 μm and a tip that measures ~26 μm with a flat surface; rather than having a radius of curvature, creating a micron-scale flat features when cutting through materials such as Kapton^®^. Looking at the top left image we can see that the cutting tool’s blade is a scalene tetragonal pyramidally shaped tool, with ellipsoidal cutting faces with a cutting angle of ~42° and 90° degree flat tip as depicted clearly in Figure 2. The cutting method utilized in this work is the drag knife method [17]. This involves dragging the blade without lifting it out of the material path. Further the path direction is depicted in Appendix A. The preset cutting forces as per the Silhoutte cutter ranges from F1–F33. These were measured using the FSR and are as follows (for the ones used in this work): F1 = 0.09415 N, F10 = 0.1285 N, F22 = 0.2962 N, F24 = 0.3236 N, F26 = 0.3433 N and F33 = 0.3629 N as shown in Table 1. Experiments with all of these forces were carried to determine the best cutting conditions for Kapton (Appendix A). Increasing the force increased the chances of plastic deformation as shown in Appendix A. Forces of F1 and F10 did not allow for the blade to penetrate all the way through the substrate as shown in Appendix A, however cutting at forces F22 and above ensured through-substrate cutting as shown in Figure 3. Using the FSR method described in Materials and Methods led to the determination of a force value of 0.2962 N for cutting all the way through the substrate. Increasing the speed was shown to have a marginal increase on edge roughness (value: 5.7 μm) and thus a speed of one (corresponding to 1 mm/s) was used for all cuts [20].

### 3.2. Design to Device Measurements

After the cutting tool, the force, and velocity of cutting were analyzed for a detailed look at the cutting parameters for stencil masks. In Figure 4, a “Straight Channels” was used as the starting reference for imaging and design to device calculations. Additionally, the error for SiO_2_ stenciled patterns are mentioned with all the different designs as it was the highest of percent errors for all materials deposited. It was found that the smallest (left) pattern machined by the plotter cutter reached an average of 257.1 μm design, however we were able to define smaller than 100 μm traces (Appendix A) just not repeatably. For “single line slit cut” designs of dimensions, 2000 μm, 1000 μm, 750 μm, 500 μm, and 250 μm, the average micromachined stencil mask (n = 3) measured: 2051.2 μm, 1128.9 μm, 875.6 μm, 617 μm and 287.3 μm, respectively. These stencils were utilized in chromium deposition as described in Section 2.3. The measurement of the metal features using the stencils (Figure 5) above were: 

2038.7 μm, 1104.7 μm, 858.3 μm, 617.4 μm, and 257.1 μm, respectively. As described in Section 2.4, to demonstrate inorganic material deposition through the stencils, gelatin was cast through the “single line slit cut” designs and they measured: 1989.6 μm, 930.1 μm, 738.5 μm, 549.9 μm and 314.9 μm, respectively. Lastly, for the *E. coli* inoculation stencils the measurements were: 2209.1 μm, 1185.7 μm, 946.5 μm, 614.8 μm and 303 μm, respectively.

For the dielectric SiO_2_ deposition these values measured 2166.92 μm, 1160.68 μm, 947.06 μm, 657.07 μm, 314.11 μm, 338.99 μm with 70% error measured when depositing at 200 μm single line channels design. These values are depicted in Figure 4.

For the serpentine design in Figure 6, it was found that the smallest pattern cut by the plotter cutter reached an average of 331.7 μm, when cutting the 500 μm design. The designed values for the various serpentines were 4000 μm, 2000 μm, 1500 μm, 1000 μm and 500 μm when compared to actual stencil values of 3903.6 μm, 1896.9 μm, 1434.6 μm, 896.2 μm, and 548.5 μm, respectively. The average values for the measured serpentine patterns during chromium deposition were: 3953.1 μm, 1976.5 μm, 1498 μm, 1043.6 μm and 589.3 μm. For *E. coli* inoculation the measured numbers were: 3671.6 μm, 1742.7 μm, 1310.2 μm, 946.8 μm and 331.7 μm, respectively. For Gelatin the results were 4023.3 μm, 2107.9 μm, 1687.9 μm, 1177.9 μm, and 596.5 μm, respectively. The dielectric SiO_2_ transitioned to 3761.2 μm, 1731.8 μm, 1260.9 μm, 870.6 μm, and 447.6 μm with the greatest percent error for SiO_2_ being 15.9%.

For the Square Channel pattern in Figure 7, it was found that the smallest pattern cut by the plotter cutter reached an average of 368.9 μm, when cutting the 500 μm design. The designed value of 250 μm was unable to be resolved. The average actual measured values from designs were: (design) 2000 μm, 1000 μm, 750 μm, 500 μm and 250 μm to (measured value) 1817.2 μm, 841.9 μm, and 643.2 μm, respectively. For Chromium Deposition, the numbers were: 1948.4 μm, 904.9 μm, 596.1 μm, 368.9 μm and for *E. coli* inoculation the numbers were: 2006.9 μm, 800.4 μm, 622.6 μm and 368.9 μm, respectively. For Gelatin this translated to 1928.0 μm, 958.1 μm, 730.8 μm, 479.3 μm, 392.4 μm and 308.9 μm. For the dielectric SiO_2_ this corresponded to 1815.8 μm, 807.5 μm, 556.1 μm, 419.6 μm, 530.5 μm and 476.0 μm with 58.7% being the largest error.

### 3.3. Interdigitated Electrodes Fabrication 

The low-cost parallel planar interdigitated electrodes with Cr/Au layers were fabricated as discussed in Section 2.5. Figure 8 depicts the full spectrum impedance performance of the devices over the frequency range of 10 Hz–100 kHz. Such an impedance measure (real part of the impedance plotted in Figure 8) and its corresponding values of Cell Index can shine light on morphology and spread of living tissues such as bacteria over a wide range of physical conditions [37,38,39].

For both devices, finger width of 1 and 2 mm behave as expected showing interdigitated type signature and reduced impedance as the pitch (or spacing) is reduced and width is increased [37,40]. The ratio of the pitch difference in the range of values is a key indication of the proportion of the electrodes, number of fingers and based upon the electrolyte solution used [38]. At low frequencies the total impedance essentially becomes resistive. The sensitivity increases by increasing the area of the contact surface between the electrodes and the sample under test. A high impedance would result in a large, applied electrode voltage leading to undesirable electrochemical reactions that may be harmful to cellular cultures (not shown here).

### 3.4. Silicon Dioxide Deposition Results

In Figure 9 we see the amorphous and polycrystalline structures of SiO_2_, and a dielectric IDE design showcased in the optical micrograph on the left. The different deposited architectures of SiO_2_ occur based on the different temperatures of deposition, at higher temperature leads to greater absorbance due to much smother substrates [41,42]. We expect that at a lower temperature of 50° Celsius, the PECVD deposition structure to be similar to those deposited using ICP PECVD SiO_2_ matrixes at these temperatures [43].

### 3.5. Optimized Kirby Bauer Stencil Mask 

On a disc diffusion assay, antibiotics slowly diffuse outwards from the impregnated discs toward the surface of the agar, creating a concentration gradient of the compound. Due to this effect, the zone of inhibition is the representation of the area containing antibiotics over the minimum concentration necessary to prevent bacterial growth. According to Bonev et al. the concentration of antibiotics decreases logarithmically from the discs [32]. Therefore, the similar inhibition length obtained from the DDT and SDDT demonstrates that the Kapton stencil does not change antibiotic diffusion as shown in Figure 10 and Figure 11.

#### 3.5.1. Length of Inhibition

As customary for DDT, the diameter of the zone of inhibition was measured with an electronic caliper, but the length from the disc to the edge of the inhibition zone was utilized to compare results with SDDT (Appendix A). The results show similar inhibition lengths for both test variations (Figure 10 and Figure 11). For *E. coli*, increasing the initial antibiotic mass results in a logarithmic increase of the inhibition length/radius. Interestingly, tests on *X. alfalfae* display a quadratic relation between antibiotic mass and inhibition length. This disparity might be due to the difference in bacteria, incubation, agar, and temperature between the tests, as it is expected these variables will have an impact on the diffusion of the compound [32].

#### 3.5.2. Area of Inhibition

In Figure 12 we show the theoretical/qualitative framework behind the transition and comparison of the Kirby Bauer and Kapton Stencil Mask. Due to antibiotic diffusion from the disc, the Minimal Inhibitory Concentration is measured at the boundary of bacterial growth and antibiotic diffusion. A regular area of inhibition is key to interpreting the DDT correctly and must not have any interruptions so that its diameter is the same in all directions. The SDDT possesses 2 mm wide channels for bacterial growth which reduces ambiguity. Based on the measurements the inhibition zones were estimated. For the DDT the area of inhibition was calculated as a perfect circle: $A=\frac{1}{4}\pi d^{2}$, where *d* is the diameter of the zone of inhibition. For the SDDT the inhibition zone was determined as an exact rectangle combined with the area of the antibiotic disc: $A=2\cdot l+28.27\;{\mathrm{mm}}^{2}$, where *l* is the length of inhibition, depictions of the Kirby Bauer DDT and SDDT are shown in Appendix A. The results show that due to the stencil the area of inhibition drastically decreases compared to the regular DDT.

For the governing equations of the SDDT test compared to the DDT correlated with each other for *E. coli* K-12 and *X. alfalfae*.

*E. coli* K-12
(1)$$y=Cln\left(x\right)+B$$

*X. alfalfae*.
(2)$$y=C_{1}X^{2}+C_{2}X+B$$

The equations governing the human *E. coli* K-12 followed the logarithmic equation predicted by Bonev [32]. The plant bacteria which *X. alfalfae* took on the form of a 2nd order polynomial, which to our knowledge has not been studied and requires further study.

A few circumstances of the traditional Kirby Bauer Test are non-favorable, with the DDT there is a limit of about 3–4 disks to be placed in a Petri dish. The circular diffusion pattern of the test can have a large overlap [44]. Our SDDT using Kapton^®^ stencils, allows for at least 6 or more tests of the same potency or lower potency as shown in Bhargav et al. [45]. The stencil mask tests are also more favorable than another antimicrobial test known as the Epsilometer-test (E-test) being 5 mm wide and 60 mm long, which contains exponential gradients with a combination of dilution and diffusion, which claims to be able to test five drugs in the same diameter plate [46,47]. This test claims to have a symmetrical inhibition ellipse centered along the strip, with the end of the strip having the largest antibiotic concentration showing the maximum distance for MIC and lowest MIC on the other side. However, both claims do not occur frequently, and the ellipses are not always uniform requiring the test to be performed repeatedly [48,49,50]. The E-test is also extremely cost prohibitive when compared to the DDT and SDDT test at >$400 for 100 tests.

The reduction of the area needed to perform the test is beneficial because more assays can be performed on a single agar plate, decreasing the resources and bacteria needed for large susceptibility screenings. Based on these findings, it is envisioned that plotter cutter-generated stencils will be utilized in the future to produce economical standardized antibiotic testing kits.

## 4. Conclusions

We have successfully shown that an inexpensive plotter cutter can be used to design rapid microstructure prototypes to full-blown applications within seconds using 300HN Kapton^®^ films. We found that these films were best suited for our application as they are thin enough to cut through in a single pass and prevent tearing while cutting. The effective parameters for cutting through the Kapton^®^ films were determined as: F = 22–26 (which was determined to be 0.2962, 0.3236, 0.3433 N); Speed= 1 mm/s with fabrication time in seconds. Design to device measurements showcase an average error of 9.66% without SiO_2_ results, with expected trends with smaller dimensions patterns showing larger errors. Both metal and silicon dioxide interdigitated electrodes were successfully fabricated and tested. Lastly, a novel growth restricted diffusion test for MIC detection of both human and plant pathogen bacteria were demonstrated. The data for the bacteria SDDT correlated with the DDT, inferring that using this SDDT will allow for a high throughput method allowing for 3 times the number of tests to be allowed in an agar Petri dish for antimicrobial characterization. Using this new protocol, we can additionally test smaller antibiotic concentrations and multiple antibiotics in the same Petri dish for assessment. Overall, the methodologies described in this work can be utilized in makerspaces, limited resource countries and laboratories without access to microfabrication facilities. The plotter cutter can rapidly design and microfabricate stencil mask prototypes for the deposition of various inorganic and organic materials and create reproducible bacterial susceptibility tests with correlation coefficients of R ≈ 0.96 and 0.985.

## Figures and Tables

**Figure 1 micromachines-14-00014-f001:**
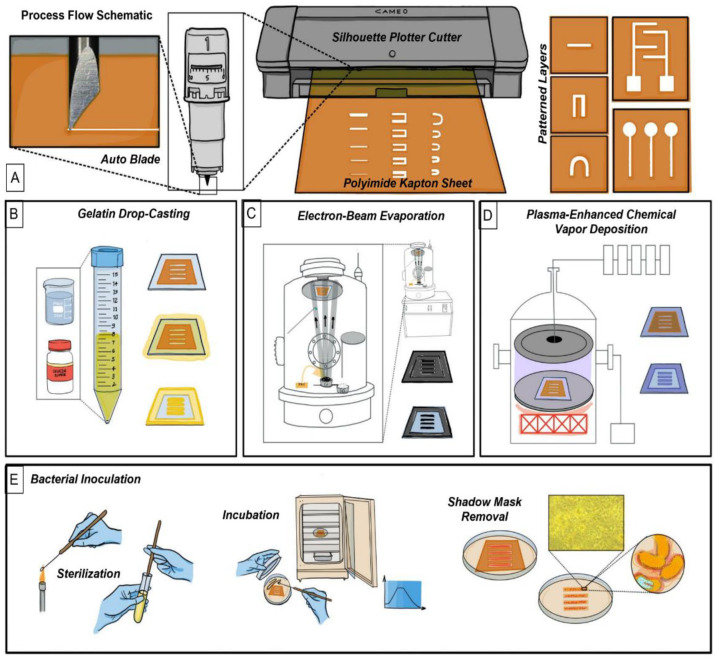
Process flow schematic: (**A**) Silhouette Plotter Cutter along with fabricated designs that were evaluated in this study (right). (**B**) Schematic of the casting process for 10 wt% gelatin. (**C**) E-beam evaporation process. (**D**) Plasma Enhanced Chemical Vapor Deposition (PECVD) process for the deposition of SiO_2_. (**E**) Culturing of *E. coli* K-12 & *X. alfalfae*.

**Figure 2 micromachines-14-00014-f002:**
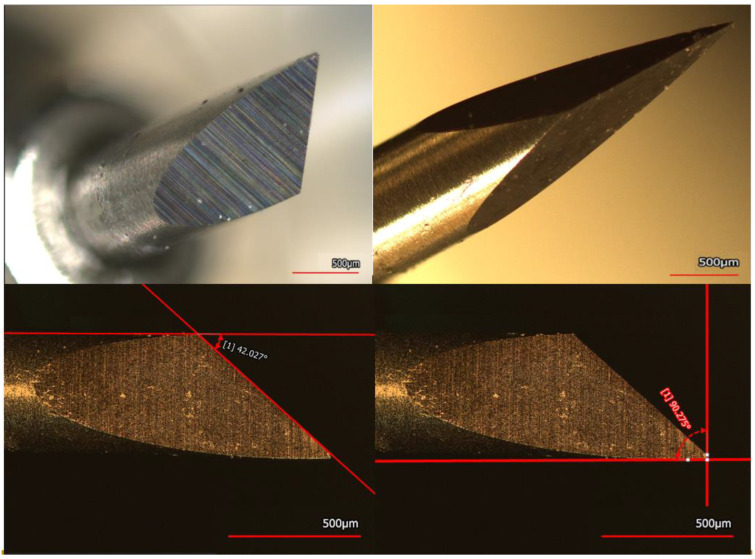
Laser confocal scanning images of the AutoBlade from the plotter cutter showcasing ellipsoidal curvature on the left and tip dimensions on the right. The blade characterization helps with assessment of cutting features on specific materials.

**Figure 3 micromachines-14-00014-f003:**
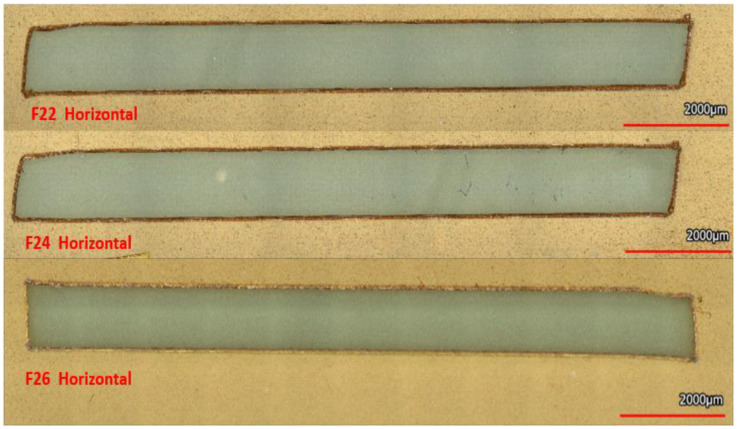
3D confocal images images of “Good Cuts” depicting F22, F24, F26 Horizontally. The Cuts are 2 mm wide and 12.7 mm in length.

**Figure 4 micromachines-14-00014-f004:**
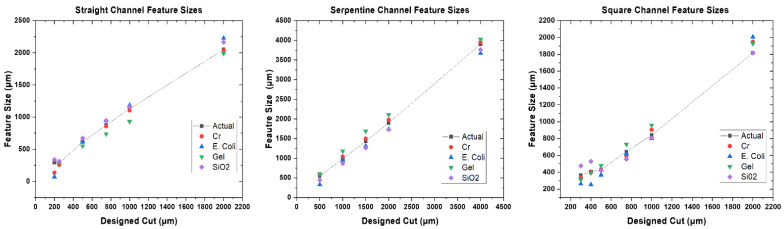
Design to device for: (**Left**) Straight Channels, (**Center**) Serpentine Channels, (**Right**) Square Channel Feature Sizes.

**Figure 5 micromachines-14-00014-f005:**
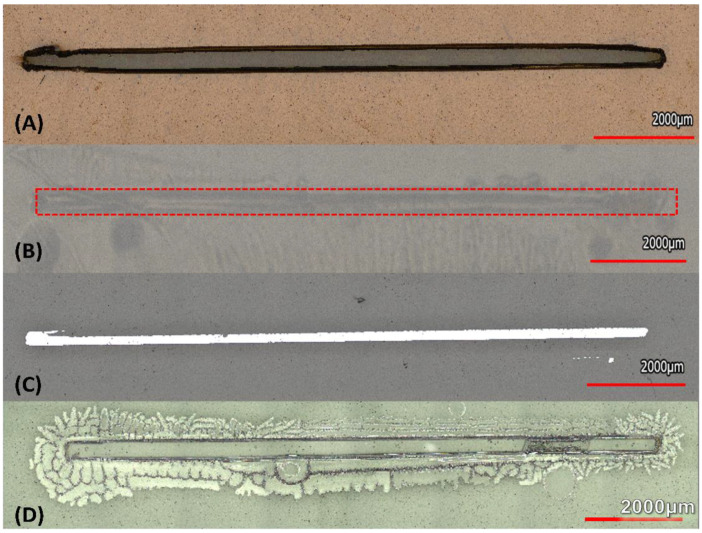
Straight Channel Cuts depicting the smallest cuts for (**A**) Stencil Mask, (**B**) *E. coli* (**C**) Cr, (**D**) Gelatin. Inconsistent Sub 100 μm gelatin structures shown in Appendix A.

**Figure 6 micromachines-14-00014-f006:**
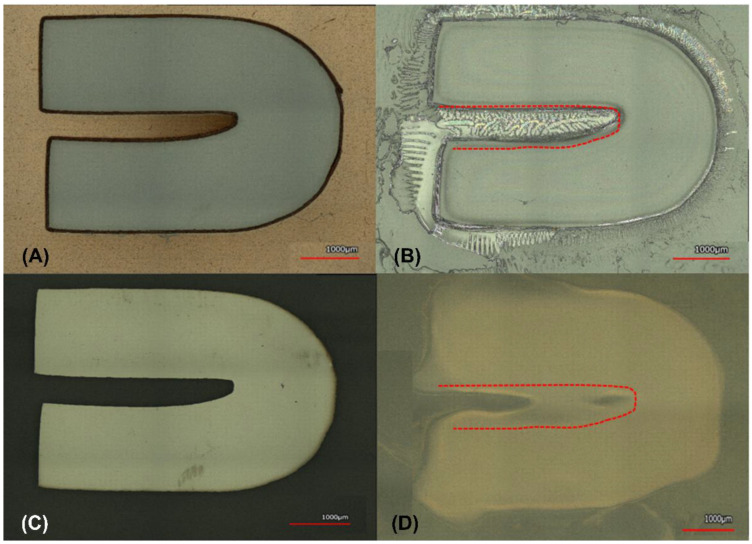
Serpentine Channel Depicting the Smallest feature sizes for (**A**) Stencil Mask, (**B**) Gelatin, (**C**) Cr, (**D**) *E. coli*.

**Figure 7 micromachines-14-00014-f007:**
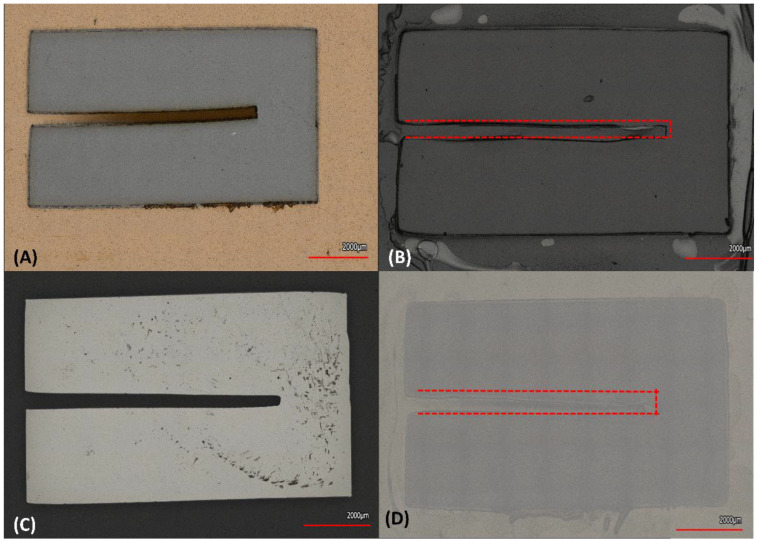
Square Channels depicting the smallest feature sizes for (**A**) Stencil Mask, (**B**) Gelatin, (**C**) Cr, (**D**) *E. coli*.

**Figure 8 micromachines-14-00014-f008:**
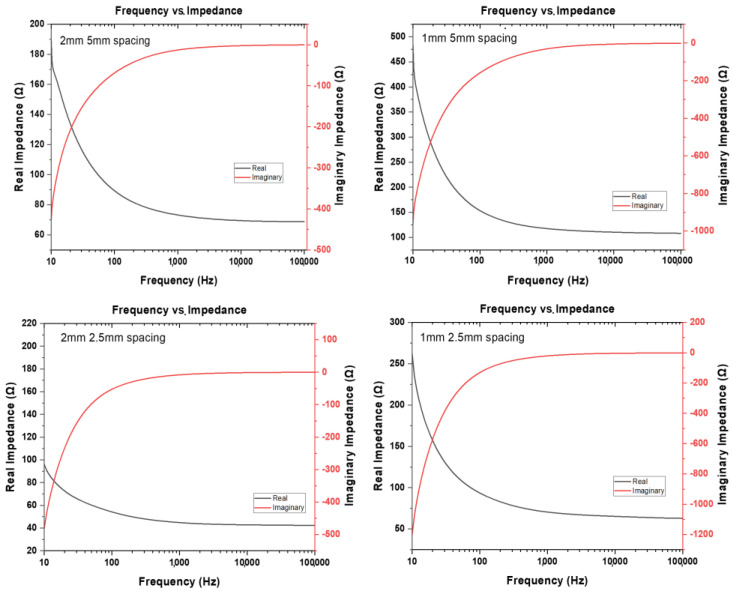
Full spectrum (Range from 1 Hz to 100 kHz) measurements for 1 mm and 2 mm pitch IDE device. (Showcased as metal patterned Cr/Au). Each device was recorded three times and a statistical average of the real and imaginary parts of the impedance are presented.

**Figure 9 micromachines-14-00014-f009:**
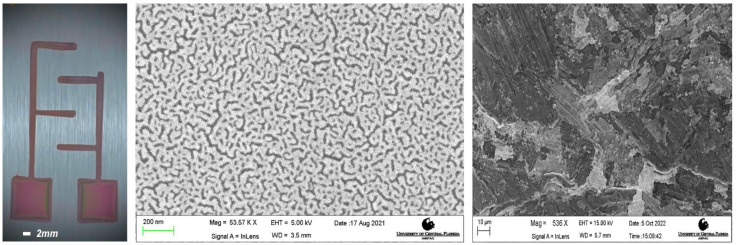
Deposition of a PECVD; (**Left**) Dielectric Interdigitated electrode, (**Center**) Deposition at 300 °C and (**Right**) Deposition at 50 °C.

**Figure 10 micromachines-14-00014-f010:**
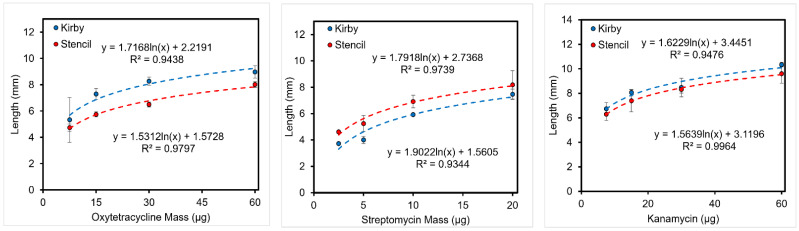
*E. coli* K-12 Antibiotic Resistance Test. (**Left**) Oxytetracycline, (**Center**) Streptomycin, (**Right**) Kanamycin.

**Figure 11 micromachines-14-00014-f011:**
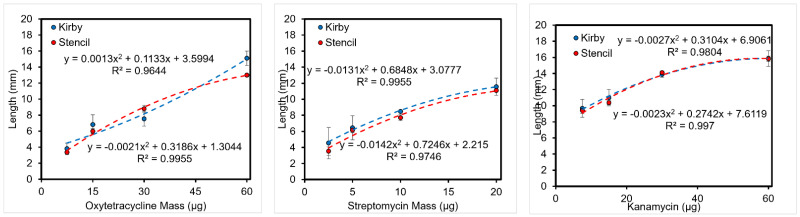
*X. alfalfae*. Plant Bacteria Antibiotic Resistance Test. (**Left**) Oxytetracycline, (**Center**) Streptomycin, (**Right**) Kanamycin.

**Figure 12 micromachines-14-00014-f012:**
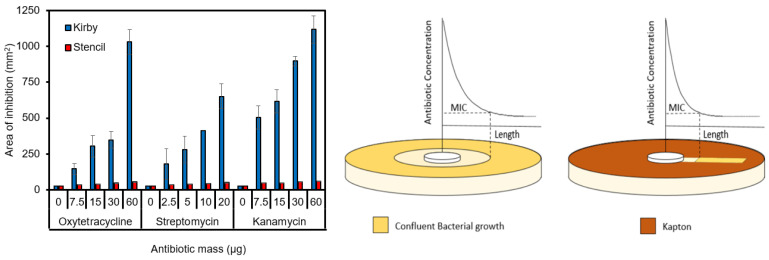
Theoretical MIC Justification. (**Left**) The change in size requirement for the area of inhibition, (**Right**) depiction of the required surface area of the bacteria growing in a Petri dish.

**Table 1 micromachines-14-00014-t001:** Characterization of Forces for Cutting Kapton.

Force	FSR (N)	Corners H Angles (°)	Corners V Angles (°)
F1	0.09415	~	~
F10	0.1285	~	~
F22	0.2962	92.48 ± 2.86	93.72 ± 1.02
F24	0.3236	93.32 ± 1.98	95.44 ± 0.97
F26	0.3433	91.18 ± 1.31	94.18 ± 1.31
F33	0.3629	~	~

Blade Depth of 10, Speed 1 and Pass 1 was used for all forces.

## Data Availability

The data of visualization and characterization and experimental tests can be sent on request. The files of Images and samples can be found in Dr. Swaminathan Rajaraman’s Laboratory in the sample box that says Andre Childs.

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
