# Peer review of "Plotter Cut Stencil Masks for the Deposition of Organic and Inorganic Materials and a New Rapid, Cost Effective Technique for Antimicrobial Evaluations"

_micromachines, 2022, doi:10.3390/mi14010014_

Round 1

Reviewer 1 Report

In this manuscript, the authors developed a facile, low-cost and easily accessible method to fabricate stencil masks by plotter-cutters. It can be accepted for publication in the present form. 

Author Response

Thank you very much for your input. Attached are the adjustments that were made by your suggestions. 

Reviewer 2 Report

This is an interesting manuscript. I have some general questions. Does the lower accuracy of the geometrical patterns in the Kapton Films has any disadvantages compared to the traditional stencil masks manufacturing? How is the Kapton Film attached on the glass slides? Was there an adhesive layer on the Kapton Film or how does it work?  What is the material of the cutting blade and what is the lifetime of it?

Does the described SDDT method have disadvantages compared to the traditional DDT method?

The caption of the figures 6-8 could be improved, if each sub figure is additional labeled with a, b, c, and d. For example, for me it is not clear where the Cr-figure is in figure 6 because the figures are inconsistent regarding the exposure settings in relation to figure 7 and 8.

There is a typo in the caption of figure 9. I think it should be given “10Hz to 100kHz”.

Author Response

Thank you very much. Attached are the adjustments to your suggestions.

Reviewer 3 Report

 In this manuscript, the authors demonstrated an inexpensive cutter-plotter for rapid fabrication of micro designs using Kapton films. They have optimized main parameters (such as force and speed) required for effective fabrication. Further, authors have performed Disk Diffusion Test for bacteria affecting both plants and humans with different antibiotics. The work is interesting and can serve the purpose of frugal fabrication systems especially in resource-constrained settings and will facilitate 1$ microfluidics approach. Authors are suggested to include a table summarizing different cutter plotter-based fabrication methods with channel type, resolution, pros & cons as well as application investigated. 

Author Response

Thank you very much. Attached is our response and adjustments to your comments. 

Reviewer 4 Report

Manuscript Number:  micromachines-2070386

Title: RAPID, COST-EFFECTIVE, PLOTTER CUT STENCIL MASKS FOR THE DEPOSITION OF ORGANIC AND INORGANIC MATERIALS AND A NEW STENCIL BASED ANTIMICROBIAL EFFICACY TESTING PLATFORM

Comments for the authors

The authors describe a plotter cutting method to cut Kapton film and use the design to investigate growth restricted diffusion test for MIC detection of both human and plant pathogen bacteria. The paper present few good insights, however, there were some major considerations to be addressed before its publication. 

Major Comments:

1.       The overall language of the manuscript needs substantial revision as there were many grammatical/language or editing mistakes within the paper.

2.       It is described that this method allows “three times the amount of antibiotics to be tested within the same nutrient agar petri dish” in the abstract. However, in the paper, this quantification is missing in the results section.

3.       Why the Kapton tape is used? The novelty behind its selection and the fabrication technique used require background and explanation. Additionally, a clear novelty statement is missing both in the abstract and the introduction.

4.       It is unclear as to how the smaller area cut by the plotter causes a better testing than the standard protocol. Can any other patterning technique do the same or is it limited to only this particular technique? For instance, we can use CO2 laser cutting to make even smaller pattern designs. Will it perform better in the antibiotic test?

Minor Comments:

1.       Some figures shown as failed attempts can be moved to supplementary information.

2.       The methodology section is also unclear. Details related to patterning and which experiments is being used for gelatin, SiO2 deposition, electrode design, and bacterial handling need to be clearly written.

3.       The title of the paper is too long.

Author Response

Comments were the same as Reviewer 3

Round 2

Reviewer 4 Report

The authors have addressed the concerns I recommend this for publication.